# Investigation of the Pathogenesis of Liver Fibrosis Associated with Type 2 Diabetes Mellitus via Bioinformatic Analysis

**DOI:** 10.3390/biomedicines13040840

**Published:** 2025-04-01

**Authors:** Zhiyu Xiong, Kan Shu, Yingan Jiang

**Affiliations:** Department of Infectious Diseases, Renmin Hospital of Wuhan University, Wuhan 430060, China; 2022203020019@whu.edu.cn (Z.X.); 2023283020094@whu.edu.cn (K.S.)

**Keywords:** liver fibrosis, T2DM, bioinformatics, machine learning, drugs

## Abstract

**Background:** The global prevalence of type 2 diabetes mellitus (T2DM) with liver fibrosis is rising, with T2DM identified as an independent risk factor and key prognostic factor for liver fibrosis. However, the underlying mechanisms remain unclear. **Methods:** To explore the shared pathogenesis of liver fibrosis and T2DM, we analyzed gene expression profiles from the GEO database. The co-differentially expressed genes (co-DEGs) were identified and subsequently analyzed through functional enrichment, protein–protein interaction (PPI) network construction, transcription factor prediction, and drug prediction. Machine learning algorithms were then applied to identify key genes. **Results:** A total of 175 co-DEGs were identified. Functional enrichment analysis indicated their involvement in extracellular matrix (ECM) remodeling, inflammation, and the PI3K/Akt signaling pathway. Through PPI network analysis and four algorithms, eight hub genes were identified, including *SPARC*, *COL4A2*, *THBS1*, *LUM*, *TIMP3*, *COL3A1*, *IGFBP7*, and *FSTL1*, with *THBS1* being recognized as a key gene by machine learning. The upregulation of *THBS1* was observed in both diseases, and it is closely related to the progression of liver fibrosis and T2DM. Transcription factor analysis detected 29 regulators of these hub genes. Drug prediction analysis suggested that retinoic acid may serve as a potential therapeutic agent. **Conclusions**: This study provides novel insights into the shared pathogenesis of liver fibrosis and T2DM and offer potential targets for clinical intervention.

## 1. Introduction

Liver fibrosis is an end-stage liver disease caused by multiple hepatotoxic factors and biliary tract diseases [1]. Chronic liver injury damages liver epithelial cells and triggers fibrogenesis, thereby inducing the activation of hepatic stellate cells (HSCs) and producing a large amount of extracellular matrix (ECM), eventually leading to tissue structure disorder and the formation of diffuse hepatic damage, ultimately causing liver dysfunction [2]. Notably, the progression of liver fibrosis and the development of type 2 diabetes mellitus (T2DM) are connected [3]. Grancini et al. showed that 35.8% of patients with liver fibrosis had impaired glucose tolerance and 48.6% of those patients had T2DM as a comorbidity [4].

T2DM is a metabolic disorder characterized by hyperglycemia due to insulin resistance [5]. T2DM is an important risk factor that promotes the development of liver fibrosis [6]. According to six cohort studies conducted by Dyal et al., patients with liver fibrosis associated with T2DM have an increased risk of developing advanced liver fibrosis (odds ratio (OR) value, 2.25–9.24) [7]. Furthermore, T2DM may also decrease patient survival rate by increasing the risk of cancer in patients with liver fibrosis associated with T2DM. Nishida found that the 5-year survival rate of patients with liver fibrosis who exhibited normoglycemia was 94.7%, whereas the 5-year survival rate of patients with liver fibrosis associated with T2DM declined to 56.6% [8]. Marengo et al. demonstrated that compared with the normal-glucose-metabolism fibrosis group, in the T2DM fibrosis group, the incidence of liver cancer was three-fold [9].

Liver fibrosis and T2DM are common chronic diseases that are interrelated. The development of liver fibrosis causes dysfunction in the receptors of the liver and pancreas, which leads to decreased insulin binding, resulting in insulin resistance and therefore the development of T2DM [10]. Cytokines may act as an important factor affecting the co-occurrence of the two diseases. An increase in the level of tumor necrosis factor α (TNF-α) may further aggravate the inflammation that contributes to liver damage [11]. Insulin resistance induces the production and activation of TNF-α, resulting in the massive release of insulin and insulin-like growth factors (IGFs) [12]. Increased production and secretion of IGF-1 may promote the proliferation of liver cancer cells and reduce their apoptosis, thereby accelerating the carcinogenesis of hepatic cells in liver fibrosis [13]. Furthermore, hyperglycemia-induced inflammatory mediators increase oxidative stress and activate HSCs in the liver, causing mitochondrial damage, lipid peroxidation, and liver fibrosis progression [14]. Additionally, the advanced glycation end products–receptor for advanced glycation end products (AGE–RAGE) signaling pathway may be one of the critical pathways involved in liver fibrosis associated with T2DM [15]. The accumulation of AGEs may further activate nuclear factor kappa B (NF-κB) signaling and other signaling pathways, thereby potentially influencing the redox balance in the hepatic cells and reactive oxygen species (ROS), which may accelerate the development of liver fibrosis [16].

In summary, compared to patients with liver fibrosis, the prognosis and treatment of patients with liver fibrosis associated with T2DM require more attention. The association between liver fibrosis and T2DM has become a matter of concern because of the increased prevalence of their co-occurrence. However, there are limited studies on the common pathogenesis of the two diseases. Therefore, the association between the two diseases and the identification of potential treatment options require further investigation.

Bioinformatic analysis has become an important method in the field of data integration [17]. In this study, the common hub genes associated with liver fibrosis and T2DM were obtained via the R program (version 3.6.3) and Cytoscape software (version 3.10.1). Based on the information obtained regarding the hub genes, the common molecular pathogenesis and potential therapeutic drugs were elucidated. These results provide new insights into the common pathogenesis of liver fibrosis and T2DM.

## 2. Materials and Methods

### 2.1. Collection of Data

This study is based on Gene Expression Omnibus (GEO) datasets (http://www.ncbi.nlm.nih.gov/geo), accessed on 12 February 2024, which are used for relevant data collection [18]. Two datasets were obtained by using “liver fibrosis” and “T2DM” as the Medical Subject Headings (MeSH) terms. The two datasets, GSE14323 and GSE29221, were selected and analyzed based on GEO datasets for this study. GSE14323 included data on 12 normal liver tissue samples from normal subjects and 12 liver tissue samples from patients with liver fibrosis. GSE29221 included data on 12 normal bone tissue samples from normal subjects and 12 skeletal muscle tissue samples from patients with T2DM.

### 2.2. Identification of Differentially Expressed Genes (DEGs)

The “Limma package” of R language is used to identify DEGs in tissue samples obtained from normal and diseased patient groups [19]. In this study, *p* < 0.05 and |log FC| > 1 were used to screen the DEGs. Thereafter, the intersection between the two gene sets was formed via the online Venn drawing website (http://bioinformatics.psb.ugent.be/webtools/Venn/), accessed on 12 February 2024, to simultaneously obtain co-DEGs involved in liver fibrosis and T2DM. Furthermore, a volcano map was created using the R program (version 3.6.3), and the DEGs with the logFC > 1 were labeled as upregulated, whereas the DEGs with the logFC < −1 were labeled as downregulated.

### 2.3. Functional Enrichment Analyses of DEGs

Functional enrichment analyses were performed via the Database for Annotation, Visualization, and Integrated Discovery (DAVID) (https://david.ncifcrf.gov/home.jsp) website, accessed on 12 February 2024, with Gene Ontology (GO) and Kyoto Encyclopedia of Genes and Genomes (KEGG) analyses used in combination. GO is a database of biological processes (BP), molecular functions(MF), and cellular components(CC) [20]. KEGG is a database dedicated to the analysis of gene-related pathways [21]. Finally, a visualization analysis was conducted using the “ggplot package” in the R program (version 3.6.3).

### 2.4. Construction of Protein–Protein Interaction (PPI) Network and Analysis of Gene Modules

The PPI network is based on the Search Tool for the Retrieval of Interacting Genes/Proteins (STRING) (https://string-db.org/), accessed on 12 February 2024, which retrieves interacting genes [22]. Subsequently, the PPI networks were established from the datasets on DEGs via Cytoscape (version 3.10.1) (http://www.cytoscape.org), accessed on 12 February 2024, with interaction scores greater than medium confidence (0.400) [23]. Meanwhile, the plug-in molecular complex detection technology (MCODE) of the Cytoscape was used to analyze key functional gene modules, with all parameters set to default values. Thereafter, the gene modules were analyzed via GO and KEGG pathway enrichment analyses.

### 2.5. Selection and Validation of Hub Genes

The Cytoscape was used to screen and select 30 hub genes that have the same expression levels in each of the five algorithms: the betweenness centrality (BC), MCC, Degree, EPC, and DMNC algorithms of the Cytoscape plug-in [24]. Thereafter, the intersection was formed using the Venn diagram to select final hub genes, which were analyzed using the KEGG analyses. Subsequently, we applied machine learning algorithms, including LASSO logistic regression and SVM-RFE algorithm, to identify the key genes [25]. Finally, the T2DM dataset GSE55605 and the merged liver fibrosis datasets GSE139602 and GSE14323 were used as the verification datasets to verify the key genes.

### 2.6. Prediction of Transcription Factors (TFs)

TFs may be useful to investigate the effects of the expression of hub genes in vivo, either physiologically or pathologically. NetworkAnalyst (https://www.networkanalyst.ca/), accessed on 12 February 2024, was used to predict the TFs capable of regulating the hub genes and to construct an interaction network for the hub genes [26].

### 2.7. Prediction of Drugs

By using Drug Signatures DataBase (DSigDB) (http://dsigdb.tanlab.org/DSigDBv1.0/), accessed on 12 February 2024, targeted drugs with high potential associated with the hub genes were predicted [27]. The DSigDB was accessed via the Enrichr platform, the data on hub genes were uploaded, and candidate drug data were obtained with *p* < 0.05 as a screening standard. Finally, the top 20 drug molecules were selected in sequence from lesser to greater *p* values.

## 3. Results

### 3.1. Identification of DEGs

Microarray gene expression datasets GSE14323 and GSE29221 include data on the samples from patients with liver fibrosis and T2DM, respectively. Based on GSE14323, 1088 DEGs were identified; based on GSE29221, 1203 DEGs were identified (Figure 1A,B). Thereafter, the Venn diagram crossover was obtained, which yielded 175 co-DEGs (Figure 1C).

### 3.2. GO and KEGG Pathway Enrichment Analyses

The DAVID was used to analyze 175 co-DEGs derived from the GO and KEGG pathway enrichment analyses. In total, 106 co-DEGs based on BP were screened, and pathways associated with 16 co-DEGs, with *p* < 0.05 as the criterion, were identified using the KEGG analysis. The KEGG pathway showed that the genes were mainly involved in four main pathways: ECM organization, protein digestion and absorption, focal adhesions, and proteoglycans in cancer (Figure 2A). The results of the GO analysis revealed that these genes were mainly involved in cell adhesion, inflammatory response, and collagen fibril organization (Figure 2B). 

### 3.3. Construction of PPI Network

Based on the 175 co-DEGs obtained, a PPI visualization network was created using STRING and Cytoscape, with a combined score of >0.4. A total of 134 nodes were involved in the grid. Subsequently, the hub genes were screened based on the BC developed by the CytoNCA (Figure 3A). The high correlation sub-network of MCODE was selected by the Cytoscape plug-in. Three closely related gene modules were obtained (Figure 3B–D). Functional enrichment analyses of these gene modules were then conducted. The GO analysis showed that the gene modules were mainly enriched in ECM organization, cell adhesion and cellular metabolism (Figure 4A). The KEGG analysis revealed that these genes were involved in focal adhesion, PI3K-AKT signaling pathways and protein absorption (Figure 4B).

### 3.4. Selection and Functional Enrichment Analyses of Hub Genes

The top 30 gene candidates were screened by employing five methods of topological analysis from the CytoHubba plug-in (Table 1). Based on the intersection of the top 30 genes ranked by the five algorithms, eight hub genes were found to be crossed, which included *SPARC*, *COL4A2*, *THBS1*, *LUM*, *TIMP3*, *COL3A1*, *IGFBP7*, and *FSTL1* (Figure 5A). The functional enrichment analyses showed that the hub genes were involved in the proteoglycan cancer pathway, protein digestion and absorption, ECM–receptor interaction pathway, PI3K/Akt signaling pathways and focal adhesion (Figure 5B). The LASSO logistic regression and SVM-RFE algorithms, applied to the eight hub genes, identified four genes through LASSO (*THBS1*, *TIMP3*, *COL3A1*, and *IGFBP7*) and four genes through SVM-RFE (*THBS1*, *LUM*, *IGFBP7*, and *SPARC*). Ultimately, *THBS1* and *IGFBP7* were selected as key genes associated with liver fibrosis and T2DM (Figure 5C–E). The T2DM dataset GSE55605 from GEO and the merged liver fibrosis datasets GSE139602 and GSE14323 were selected as the validated datasets to verify the reliability of the expression levels of the key genes (Figure 5F). The results indicated that, compared to normal tissue samples, the expression levels of *THBS1* were upregulated in the T2DM patients and the liver fibrosis patients. In contrast, the expression levels of *IGFBP7* were upregulated in liver fibrosis patients but remained unchanged in T2DM patients.

### 3.5. Prediction of TFs

The prediction of TFs of the hub genes was performed via NetworkAnalyst. The results of the prediction showed that *THBS1*, *COL4A2*, and *TIMP3* were regulated by 29 TFs. *SMC3* was capable of simultaneously regulating *THBS1* and *COL4A2*, and *BHLHE40* was capable of simultaneously regulating *THBS1*, *COL4A2*, and *TIMP3* (Figure 6).

### 3.6. Prediction of Drugs

To screen the potential intervention drugs, the DSigDB Enrichr platform was used to search the top 20 drugs, which were screened with *p* < 0.05 as the screening standard (Table 2). The results showed that retinoic acid (CTD 00006918), cyclosporin A (CTD 00007121), and cytarabine (CTD 00005743) were the potential targeted drugs that interacted with most of the hub genes.

## 4. Discussion

The incidence of liver fibrosis associated with T2DM has been increasing. Compared to patients with liver fibrosis, patients with T2DM as a comorbidity have more rapidly progressing liver fibrosis, which may even result in hepatocellular carcinoma (HCC) in severe cases [28]. Considerably high insulin resistance is the main factor affecting the development and prognosis of liver fibrosis associated with T2DM [29]. However, there are no clinical treatments that may be considered the best therapy for patients with liver fibrosis associated with T2DM. Therefore, the investigation of the common pathogenesis of liver fibrosis and T2DM will help provide more information to identify potential targeted drugs for the treatment of these co-occurring diseases.

In this study, a combination of bioinformatic analyses was used to identify 175 co-DEGs associated with liver fibrosis and T2DM. The results of the analysis of co-DEGs suggested that the co-DEGs were predominantly enriched in the proteoglycan cancer pathway, proteolytic and absorbent protein pathway, ECM–receptor interaction pathway, PI3K/Akt signaling, and AGE–RAGE signaling pathway. PPI network and four algorithms identified eight common hub genes, including *SPARC*, *COL4A2*, *THBS1*, *LUM*, *TIMP3*, *COL3A1*, *IGFBP7*, and *FSTL1*, and further selection using LASSO and SVM-RFE highlighted *THBS1* and *IGFBP7* as key genes. Validation confirmed that *THBS1* was upregulated in both conditions, while *IGFBP7* was mainly elevated in liver fibrosis. TFs associated with hub genes form a complex interaction network to elucidate the pathogenesis of liver fibrosis associated with T2DM. Specifically, *THBS1* stands out as a particularly significant gene in this network, with its regulation by both *SMC3* and *BHLHE40* highlighting its central role in the progression of liver fibrosis and T2DM. Moreover, drug screening using the DSigDB platform identified retinoic acid, cyclosporin A, and cytarabine as potential therapeutic agents targeting the identified hub genes.

The etiology of both diseases may be associated with inflammatory cytokine response, ECM organization, and PI3K/Akt signaling. Inflammation is an important contributor to the development and progression of liver fibrosis [30]. An imbalance in the level of pro-inflammatory or anti-inflammatory factors may promote the transformation of HSCs and the activation of myofibroblasts, which contribute to accelerating the progression of liver fibrosis [31]. Inflammatory cytokines are one of the essential factors that induce T2DM by directly acting on pancreatic β-cells, thereby damaging cellular DNA [32]. Its effects may trigger oxidative stress in pancreatic β-cells, which leads to apoptosis, resulting in the generation of numerous free radicals of oxygen that induce insulin resistance [33]. P13K/Akt signaling pathway may trigger apoptosis of pancreatic β-cells and insulin resistance [34]. It may also participate in Smad3 signal transduction, thereby causing reduced insulin sensitivity and eventually affecting metabolic capacity [35]. Furthermore, the PI3K/Akt signaling pathway can also stimulate the activation of hepatic Stellate Cells (HSCs) and myofibroblasts and induce the production of ECM, thereby resulting in liver fibrosis [36].

Thrombospondin-1 (THBS1) is a glycoprotein that plays a critical role in various physiological and pathological processes [37]. *THBS1* is a major activator of transforming growth factor (TGF)-β1 and interacts with aVb3 and a2Bb3 complex receptors on HSCs to activate them, although the specific mechanism remains unclear [38]. Furthermore, *THBS1* may be associated with M1 macrophages, which exacerbate liver tissue injury by promoting inflammatory responses [39,40]. Li et al. demonstrated that upregulation of THBS1 in oral keratinocytes activates p38, Akt, and SAPK/JNK signaling pathways, which can polarize M1-like tumor-associated macrophages (TAMs) through an exosome-mediated pathway [41]. However, it remains uncertain whether *THBS1* directly induces M1 macrophages, thereby accelerating the progression of liver fibrosis. Moreover, in T2DM, *THBS1* was robustly induced by high glucose [42]. A glucose response element has been identified in the promoter region of *THBS1* and hyperglycemia is both a cause and a consequence of metabolic stress [42]. Raman et al. have reported elevated *THBS1* transcription in endothelial cells as a response to high glucose [43]. *THBS1* also leads to platelet aggregation, angiogenesis, and tumorigenesis, which may be an important risk factor for the occurrence of T2DM [44]. Therefore, *THBS1* may be a potential target gene for regulating liver fibrosis associated with type 2 diabetes.

The results of DSigDB suggest that retinoic acid (a derivative of vitamin A), cyclosporin A, and cytarabine may affect hub genes. Kang et al. discovered that retinoic acid markedly attenuates the cell toxicity induced by interleukin-1 (IL-1), interferon-γ, and other cytokines, resulting in the apoptosis of pancreatic β-cells [45]. Retinoic acid may also regulate the expression of IL-17A, a vital inflammatory factor associated with liver fibrosis [46]. However, cyclosporine A and cytarabine have been shown to have some effects on glucose metabolism, though they are not typical treatments for diabetes. Thus, retinoic acid may be the drug for the treatment of liver fibrosis associated with T2DM.

The limitations of this study include the small sample size of the validation datasets, which showed an upward trend but was also revealed a considerable difference between the trained group and the validation group. Further research is required to investigate the function of *THBS1* in vitro and in vivo. Additionally, modern antidiabetic drugs, such as GLP-1 receptor agonists (GLP-1RAs), have shown significant effects, particularly in controlling blood glucose levels, promoting weight loss, and improving metabolism [47]. Some preliminary studies suggest that GLP-1RAs may have anti-fibrotic potential, particularly in improving liver function and reducing liver damage [48]. Further research could explore the potential of combining GLP-1RAs with retinoic acid to enhance therapeutic outcomes in patients with liver fibrosis associated with diabetes. This combination approach could potentially offer synergistic effects, improving both glucose control and liver health.

## 5. Conclusions

This study utilized a series of bioinformatic analyses and conducted differential gene expression analysis on data related to liver fibrosis and T2DM. Various common etiological mechanisms are mediated by specific hub genes associated with liver fibrosis and T2DM. *THBS1* may serve as a key regulatory factor in both conditions, warranting further investigation. The use of retinoic acid as a treatment for liver fibrosis associated with T2DM should be further validated through various experiments. In conclusion, the study results provide important information for further investigation of the molecular mechanisms involved in liver fibrosis associated with T2DM.

## Figures and Tables

**Figure 1 biomedicines-13-00840-f001:**
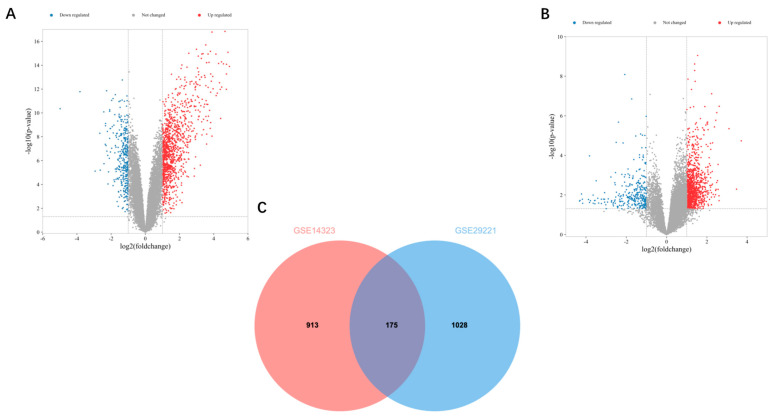
Volcano and Venn diagrams: (**A**) The volcano map of GSE14323. (**B**) The volcano map of GSE29221. Upregulated genes are marked in light red and downregulated genes are marked in light blue. (**C**) The two datasets show an overlap of 175 DEGs.

**Figure 2 biomedicines-13-00840-f002:**
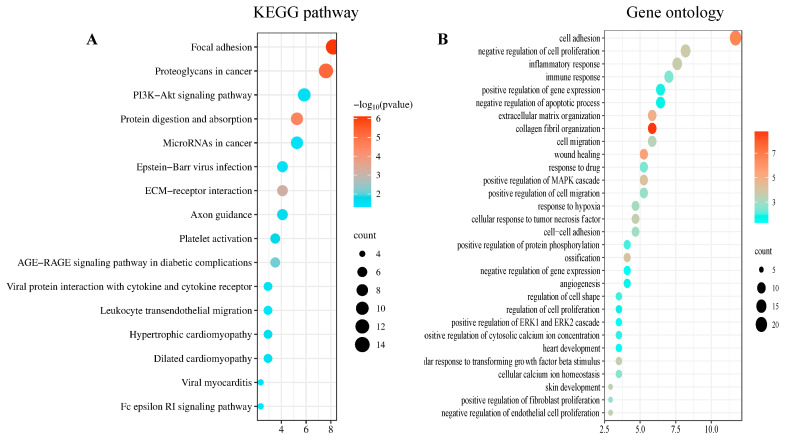
DEGs enrichment analysis results: The enrichment analysis results of (**A**) the KEGG and (**B**) the GO analyses. *p* < 0.05 was considered significant.

**Figure 3 biomedicines-13-00840-f003:**
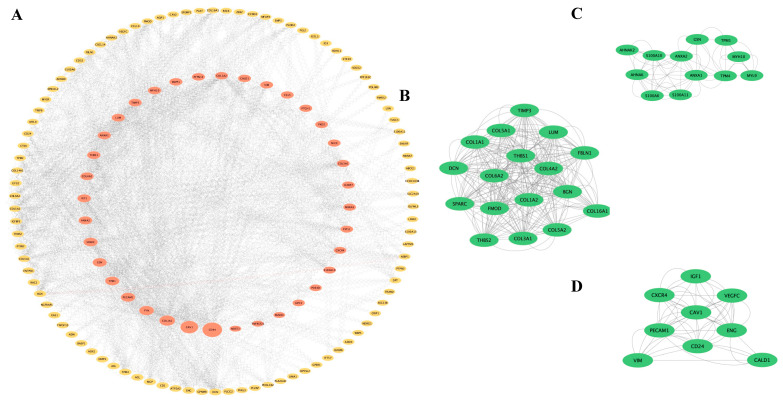
PPI network and significant gene modules: (**A**) The PPI network diagram. Red indicates genes with a high betweenness value and yellow indicates genes with a low betweenness value. (**B**–**D**) Three clustering gene modules. The green indicates significant genes.

**Figure 4 biomedicines-13-00840-f004:**
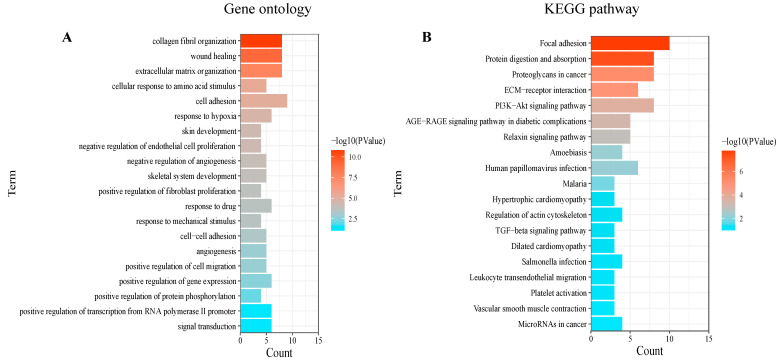
Enrichment analysis of gene modules: The enrichment analysis results of (**A**) the GO and (**B**) the KEGG analyses. *p* < 0.05 was considered significant.

**Figure 5 biomedicines-13-00840-f005:**
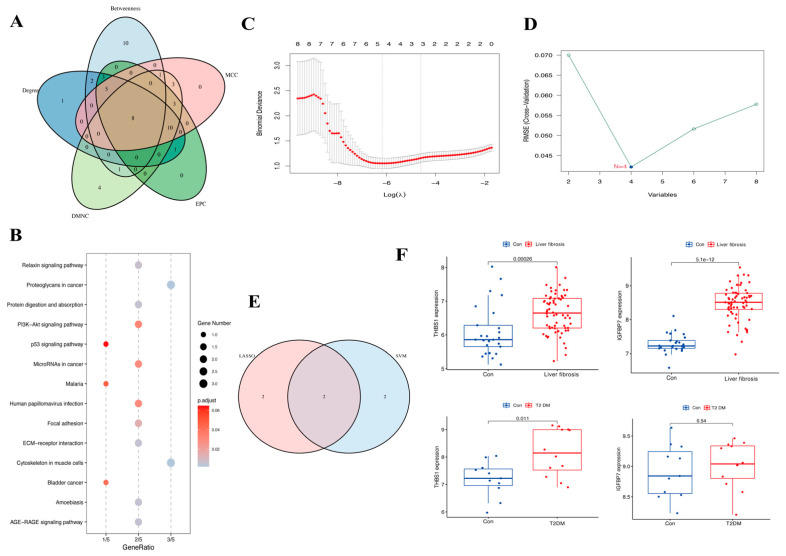
Venn diagram, enrichment analysis of hub genes, and the expression level of key genes: (**A**) The five calculated results show an overlap of eight hub genes. (**B**) Results of the KEGG enrichment analysis. (**C**,**D**) Results of LASSO logistic regression and SVM-RFE algorithms. (**E**) The Venn results of two machine learning results. (**F**) Validation of key genes in liver fibrosis-related datasets and T2DM-related datasets.

**Figure 6 biomedicines-13-00840-f006:**
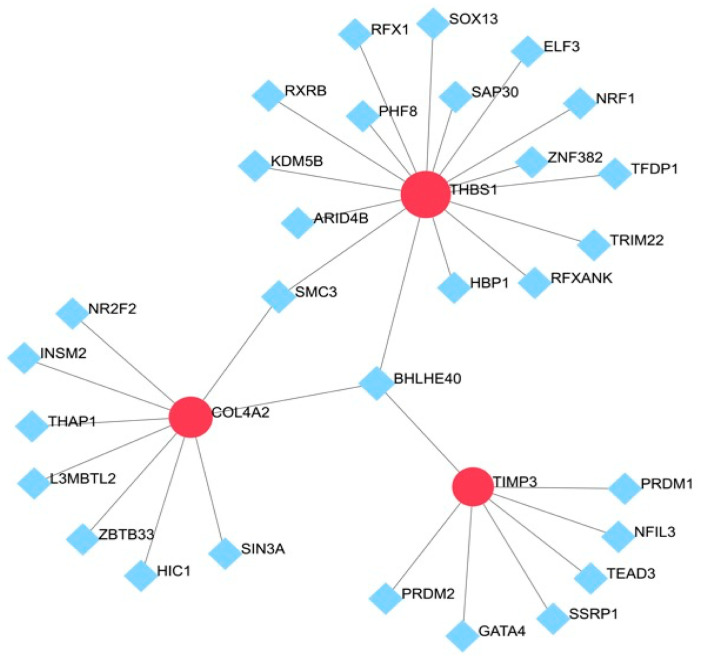
TF–gene interaction network for hub genes: The highlighted red node represents the hub genes and other nodes represent TF–gene interactions.

**Table 1 biomedicines-13-00840-t001:** Top 30 hub genes of different algorithms.

Betweenness	MCC	DMNC	EPC	Degree
CD44	COL1A1	FMOD	COL1A1	COL1A1
CAV1	COL1A2	FBLN1	COL1A2	CD44
COL1A1	BGN	DPT	COL3A1	COL1A2
FYN	COL6A2	COL14A1	LUM	COL3A1
PECAM1	LUM	MXRA5	SPARC	SPARC
TPM1	COL3A1	COL16A1	BGN	BGN
GSN	COL5A1	MFAP4	THBS1	LUM
SPARC	DCN	COL6A2	COL4A2	THBS1
ANXA2	SPARC	PLOD2	COL6A2	DCN
IGF1	TIMP3	THBS2	DCN	COL5A1
COL4A2	COL4A2	AEBP1	CD44	CAV1
THBS1	THBS1	COL4A2	COL5A2	TIMP3
ANXA1	THBS2	CTSK	COL5A1	COL4A2
LUM	FBLN1	LUM	TIMP3	COL6A2
TIMP3	COL5A2	BGN	FBLN1	COL5A2
MYH10	FMOD	COL5A2	THBS2	IGF1
ENPP1	COL14A1	MGP	FMOD	THBS2
PTPN13	COL16A1	FSTL1	COL14A1	PECAM1
COL1A2	MFAP4	PCOLCE2	IGF1	COL141
CALD1	CD44	DCN	COL16A1	FBLN1
VIM	FSTL1	IGFBP7	FSTL1	FMOD
CCL5	IGFBP7	COL5A1	MFAP4	CALD1
PTCH1	PLOD2	TIMP3	IGFBP3	ANXA2
PKD2	DPT	SPARC	CALD1	VIM
SGCE	IGF1	S100A11	CAV1	FSTL1
COL3A1	MXRA5	MYH10	IGFBP7	IGFBP3
IGFBP7	IGFBP3	IGFBP3	PECAM1	IGFBP7
MXRA5	MGP	VEGFC	VIM	ANXA1
FSTL1	AEBP1	THBS1	ENG	COL16A1
CXCR4	PECAM1	COL3A1	MGP	ENG

**Table 2 biomedicines-13-00840-t002:** Prediction of top 10 candidate drugs.

Term	*p*-Value	Genes
Cytarabine CTD 00005743	1.19 × 10^−6^	COL3A1; COL4A2; LUM; TIMP3; IGFBP7; THBS1; FSTL1
Dasatinib CTD 00004330	2.55 × 10^−5^	COL3A1; COL4A2; LUM; IGFBP7
Testosterone enanthate CTD 00000155	3.23 × 10^−5^	COL3A1; COL4A2; LUM; TIMP3
Methotrexate CTD 00006299	5.80 × 10^−5^	COL3A1; IGFBP7; THBS1; FSTL1
Retinoic acid CTD 00006918	1.29 × 10^−4^	COL3A1; COL4A2; LUM; TIMP3; IGFBP7; THBS1; FSTL1
Cyclosporin A CTD 00007121	3.00 × 10^−4^	SPARC; COL4A2; LUM; TIMP3; IGFBP7; THBS1; FSTL1
Aspirin CTD 00005447	0.00111214	COL3A1; SPARC; THBS1
Tert-butyl hydroperoxide CTD 00007349	0.00113484	COL3A1; SPARC; COL4A2; TIMP3
Arachidonic acid CTD 00007139	0.00192214	SPARC; IGFBP7
Coxistac CTD 00000539	0.00194463	TIMP3; IGFBP7

## Data Availability

In this study, all data came from publicly available databases, and references to available data are included in the methodology section.

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
