# Peer review of "Investigation of the Pathogenesis of Liver Fibrosis Associated with Type 2 Diabetes Mellitus via Bioinformatic Analysis"

_biomedicines, 2025, doi:10.3390/biomedicines13040840_

Round 1

Reviewer 1 Report

Comments and Suggestions for Authors

Very interesting paper, with high novelty and strong results! 

Could the author comment also on the applicability of this kind of analysis also on potential complications of cirrhosis, for example diabetes-induced HCC in cirrhotic patients ? (in this regard cite the comprehensive review on this topic: PMID: 23845075 )

Could this analysis consider also the potential impact of antidiabetic medications, particularly concerning the more modern GLP-1RAs ?

The discussion is quite short and it should be enriched with the concepts above

Author Response

Reviewer# 1

Very interesting paper, with high novelty and strong results! 

Comments 1:Could the author comment also on the applicability of this kind of analysis also on potential complications of cirrhosis, for example diabetes-induced HCC in cirrhotic patients? (in this regard cite the comprehensive review on this topic: PMID: 23845075 IF: 2.4 Q3 )

Author Reply1: I appreciate your valuable feedback, which has helped improve the manuscript. In response to your comment, I have added this discussion and cited the comprehensive review (PMID: 23845075 ) as you suggested to support this perspective on Page 10, Line222-225. Thank you again for your sincere comments and helpful suggestions.

Comments 2:Could this analysis consider also the potential impact of antidiabetic medications, particularly concerning the more modern GLP-1RAs ?

Author Reply2: Thanks for your professional comments! As suggested, I have included this topic in the Discussion section on Page 11, Line 289-297, where I provide an explanation of the potential effects of GLP-1RAs. Additionally, I have proposed that future studies could explore combination therapy with dual agents as a potential treatment strategy. Thank you again for your thoughtful feedback.

Comments 3:The discussion is quite short, and it should be enriched with the concepts above.

Author Reply3: Thank you for pointing out the issues of discussion section. In response, we have thoroughly revised and rewritten the whole discussion section from Page 10  to Page 11. If there are any unclear parts requiring further clarification in the revised manuscript, please let us know, and we will make the necessary revision immediately. We hope this improved version meets your expectations. Thank you again for your constructive suggestions.

Reviewer 2 Report

Comments and Suggestions for Authors

Comments and Suggestions:

Title: Investigation of the pathogenesis of liver cirrhosis associated with type 2 diabetes mellitus via bioinformatic analysis.

The manuscript by Xiong et. al., describes about the co-morbid conditions of type-2 diabetes mellitus (T2DM) and liver cirrhosis. The authors used GEO datasets and identified 175 co-differentially expressed genes (DEGs) which are involved in inflammatory cytokine response etc. They identified 8 hub genes (SPARC, COL4A2, THBS1, LUM, TIMP3, COL3A1, IGFBP7, and FSTL1) with 29 transcriptional regulators. They concluded that these insights will be beneficial for the pathogenesis of liver cirrhosis and T2DM.

The manuscript seems to be three years old and does not provide new information, but few points needs to be addressed.

Major Points:

  1. Point 2.5: the authors have used GSE7014 datasets for validating the 8 hub genes. the authors could have used other datasets which contains both liver cirrhosis and T2Dm samples.
  2. Figure 5C: Most of the hub genes did not show the similar deregulation profiles as the test datasets. How do the authors reach to the conclusion?
  3. Conclusion: the authors have mentioned that retinoic acid can be used as treatment for liver cirrhosis associated with T2DM? what about other two drugs, cyclosporin A (CTD 214 00007121), and cytarabine (CTD 00005743)? Do they also have significance with respect to these diseases?
  4. Figure 5B: the figure is difficult to understand. Please use a simple figure to show the results.

Minor Points:

  1. Line 88 and 90: please write ‘normal subjects’ in place of ‘healthy patients. How they are both heathy and patients at the same time.
  2. Point 2.1: Please mention the platforms for both datasets used. Did they used same platform?
  3. References: the references are provided till 2022. Please update references of 2023, 2024 and 2025.

Author Response

Reviewer# 2

Comments and Suggestions:

Title: Investigation of the pathogenesis of liver cirrhosis associated with type 2 diabetes mellitus via bioinformatic analysis.

The manuscript by Xiong et. al., describes about the co-morbid conditions of type-2 diabetes mellitus (T2DM) and liver cirrhosis. The authors used GEO datasets and identified 175 co-differentially expressed genes (DEGs) which are involved in inflammatory cytokine response etc. They identified 8 hub genes (SPARC, COL4A2, THBS1, LUM, TIMP3, COL3A1, IGFBP7, and FSTL1) with 29 transcriptional regulators. They concluded that these insights will be beneficial for the pathogenesis of liver cirrhosis and T2DM.

The manuscript seems to be three years old and does not provide new information, but few points need to be addressed.

Major Points:

Comments 1:Point 2.5: the authors have used GSE7014 datasets for validating the 8 hub genes. the authors could have used other datasets which contains both liver cirrhosis and T2DM samples.

Author Reply1: Thank you for your valuable feedback and suggestion. I apologize for not being able to find a dataset that includes both liver cirrhosis and T2DM samples. As an alternative, I utilized the GSE55605 dataset, which contains a larger number of T2DM-related samples, along with the GSE139602 and GSE14323 merged datasets, which are related to liver fibrosis, and used these datasets for the validation of the genes. This information has been added on Page 7, Line 191-197 to the revised manuscript for your review. Thank you again for your valuable feedback, which has greatly helped to improve the clarity of our study.

Comments 2.Figure 5C: Most of the hub genes did not show the similar deregulation profiles as the test datasets. How do the authors reach to the conclusion?

Author Reply2: Thank you for your insightful comment. After performing a more precise selection using LASSO and SVF (Figures 5C-E), we identified key genes for further validation. As shown in Figure 5F, THBS1 was upregulated in both liver fibrosis and T2DM, while IGFBP7 showed a more pronounced upregulation in liver fibrosis, with less noticeable changes in T2DM. Based on these findings, we conclude that THBS1 is likely a key gene. This gene's relevance is also discussed in the discussion section of the manuscript. We have added this information in the revised manuscript on Page 7, Line 187-190 and Page10-11, Line 261-277. Once again, we sincerely appreciate your attention to detail and your valuable suggestion. Figure 5 is shown in Word document below.

Comments 3:Conclusion: the authors have mentioned that retinoic acid can be used as treatment for liver cirrhosis associated with T2DM? what about other two drugs, cyclosporin A (CTD 214 00007121), and cytarabine (CTD 00005743)? Do they also have significance with respect to these diseases?

Author Reply3: Thank you for your thoughtful question. In the discussion section of the manuscript, we discuss the potential roles of three drugs—retinoic acid, cyclosporin A, and cytarabine—in the context of liver fibrosis associated with T2DM. Retinoic acid has shown significant effects in attenuating cell toxicity induced by interleukin-1 (IL-1), interferon-γ, and other cytokines, leading to apoptosis of pancreatic β-cells. Furthermore, it can regulate the expression of IL-17A, a critical inflammatory factor associated with liver fibrosis. Given these properties, retinoic acid is suggested as a potential treatment for liver fibrosis in the context of T2DM.While cyclosporine A and cytarabine have demonstrated some effects on glucose metabolism, they are not typically used as treatments for diabetes. Consequently, in the conclusion section, we did not specifically highlight cyclosporine A and cytarabine, as their relevance to the treatment of liver fibrosis associated with T2DM remains less prominent compared to retinoic acid. Thank you again for your insightful comment.

Comments 4: Figure 5B: the figure is difficult to understand. Please use a simple figure to show the results.

Author Reply4: Thank you for your feedback. We apologize for any confusion caused by Figure 5B. We have already presented the result of the intersection of the five algorithms in Figure 5A, where we identified the hub genes. We hope this revision makes the analysis clearer, and we appreciate your understanding. Figure 5A is shown in Word document below.

Minor Points:

Comments 5: Line 88 and 90: please write “normal subjects” in place of “healthy patients”. How they are both heathy and patients at the same time.

Author Reply5: Thank you so much for your detailed comments regarding the figures. We apologized for the unclear explanation of this. According to your suggestions, we have revised them to the latest research in the revised manuscript (Page 2, Line 83 &Line 84).

Comments 6:Point 2.1: Please mention the platforms for both datasets used. Did they used same platform?

Author Reply6: Thank you for your valuable comment. All the datasets used in our study were obtained from the GEO database. I have added the relevant information on Page 2, Line 78-79. Thank you again for your helpful feedback. If any further clarifications are necessary, please kindly let me know, and I will make the revisions promptly.

Comments 7:References: the references are provided till 2022. Please update references of 2023, 2024 and 2025.

Author Reply7: Thank you for your suggestion. We have updated all references to include the most recent literature from 2023, 2024, and 2025. You can find these updated references in the REFERENCE section from Page 12-16. We appreciate your attention to this detail.

Reviewer 3 Report

Comments and Suggestions for Authors

Thank you for the opportunity to review the article entitled "Investigation of the pathogenesis of liver cirrhosis associated with type 2 diabetes mellitus via bioinformatic analysis".

The manuscript is interesting and the results of the study are worth sharing with the scientific community and practicing physicians.

I have no reservations about the introduction or abstract, which clearly introduce the reader to the issues discussed in the rest of the article.

The methodological part also remains without reservations.

The results are presented in a clear way, easy to understand by the reader interested in the topic.

The discussion, although concise, covers all the most important aspects discussed in the literature to date.

The article does not bear the signs of plagiarism, whether partial or complete.

I congratulate the authors on their effort and wish them further success in their scientific careers.

Author Response

Reviewer# 3

Comments:Thank you for the opportunity to review the article entitled "Investigation of the pathogenesis of liver cirrhosis associated with type 2 diabetes mellitus via bioinformatic analysis".

The manuscript is interesting, and the results of the study are worth sharing with the scientific community and practicing physicians.

I have no reservations about the introduction or abstract, which clearly introduce the reader to the issues discussed in the rest of the article.

The methodological part also remains without reservations.

The results are presented in a clear way, easy to understand by the reader interested in the topic.

The discussion, although concise, covers all the most important aspects discussed in the literature to date.

The article does not bear the signs of plagiarism, whether partial or complete.

I congratulate the authors on their effort and wish them further success in their scientific careers.

Author Reply: Thank you very much for your positive feedback and recognition of our work. We sincerely appreciate your thorough review and encouraging words. Your support motivates us to continue our research in this field, and we are grateful for your valuable time and insights.

Round 2

Reviewer 1 Report

Comments and Suggestions for Authors

The manuscript is OK now. Thank you!